# DRAMNet: Depth-initialized Region-Adaptive Map Network for Single-Image Deblurring

## Abstract

Recent advances in image deblurring have achieved impressive results, yet existing methods still struggle with two key challenges: the scarcity of training data compared to other image restoration tasks and the inability to effectively handle variable blur strength across different image regions. We present DRAMNet, a three-part system that addresses these issues by transferring knowledge from the depth estimation task and using a specially designed component to assess and adapt to varying blur strength across the image. Per-patch blur map estimation allows the model to react differently to heavily and lightly blurred sub-regions, while depth information from pre-training provides structural guidance even with limited deblurring-specific data. Extensive experiments on the most popular synthetic (GoPro, REDS) and real-world (RSBlur, RealBlur) benchmarks show that DRAMNet outperforms state-of-the-art methods across the PSNR, SSIM, and LPIPS metrics. We made our code available at [Link is removed for blind review].

## 1 Introduction

Image deblurring is a fundamental challenge in computational photography with applications ranging from consumer photography to medical imaging and security systems. Despite recent advances using deep learning approaches (Fang et al., 2023; Dong et al., 2023), two significant obstacles continue to limit progress in this field.

First, unlike other image restoration tasks such as super-resolution or denoising that benefit from abundant training data (Wang et al., 2019; Ye et al., 2023; Elad et al., 2023; Li et al., 2024; Jebur et al., 2024), deblurring suffers from a significant data scarcity problem. Real-world blur datasets are limited in both quantity and diversity, while synthetic blur generation often fails to capture the complex physical processes that create natural blur patterns (Rim et al., 2022; Cao et al., 2022; Wei et al., 2022; Alutis et al., 2023). This data gap forces models to learn from relatively small or simplified datasets, limiting their generalization capabilities to real-world scenarios.

Second, even recent state-of-the-art approaches (Dong et al., 2023; Kong et al., 2023; Mao et al., 2023) treat blur uniformly across the entire image, despite the fact that real-world blur is inherently non-uniform. In a typical photograph, some regions may be heavily blurred due to object motion or depth-of-field effects, while others remain relatively sharp. Existing methods often apply the same processing algorithms to all image regions, leading to inadequate restoration in heavily blurred areas or unnecessary artifacts in regions that were already sharp.

These limitations have driven researchers to explore various strategies. Some have focused on architectural improvements, with both CNN-based (Nah et al., 2017b; Chen et al., 2021a) and Transformer-based models (Zamir et al., 2022; Wang et al., 2022b) achieving notable results on standard benchmarks. Others have focused on generative approaches (Kupyn et al., 2018) or multiscale processing (Zamir et al., 2021). However, few approaches have directly addressed the data scarcity problem or effectively handled the spatially varying nature of blur.

Notably, AdaRevD (Mao et al., 2024) introduced a patch-level blur severity classifier to adaptively allocate decoder resources. It measures the degree of blur by computing the absolute difference between the ground truth and blurred patches, then assigns each patch to one of several discrete blur levels. Based on this classification, AdaRevD dynamically determines how many decoder blocks to apply per patch, improving inference efficiency. However, this strategy is solely aimed at reducing

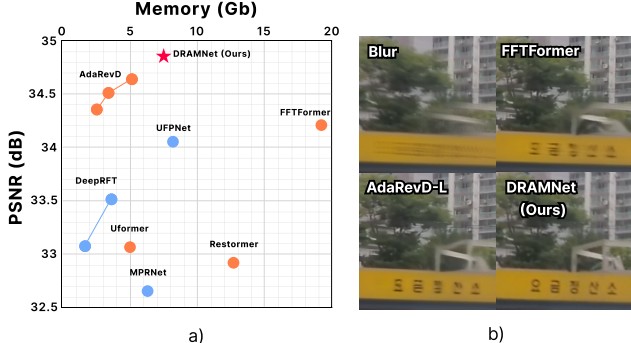

Figure 1: **(a)** PSNR *vs.* training-memory footprint on GoPro for recent deblurring networks. **(b)** Representative image crops from the same dataset. Our DRAMNet (red star) reaches the highest PSNR while requiring only moderate GPU memory. Transformer-based methods are marked in orange, CNN-based ones in blue.

computation rather than guiding the restoration process itself, and the use of GT-blur pixel difference as a blur metric may not faithfully reflect perceptual blur severity. In parallel, the Distortion-Guided Network (Purohit et al., 2021) predicts a per-pixel distortion map and switches between restoration branches, demonstrating that explicit conditioning on local blur levels can further improve quality and efficiency.

In this work, we introduce DRAMNet (Depth-initialized Region-Adaptive Map Network), a novel framework designed to address both of the aforementioned challenges simultaneously. To overcome the data scarcity issue, DRAMNet leverages transfer learning from the depth estimation domain, where large-scale datasets are rarely available. We chose the depth domain because recent works have demonstrated that depth cues are effective for video deblurring (Xu & Jia, 2012; Torres et al., 2024). Instead of explicitly designing depth-aware architectural modules, we extract structural cues by leveraging a pretrained encoder.

By pre-training on the Depth-Anything-v2 dataset (Yang et al., 2024) and transferring this knowledge to the deblurring task, our model gains robust structural priors that generalize well even with limited deblurring-specific data. Figure 1 provides qualitative and quantitative results that demonstrate the efficiency of the proposed approach over the previous state-of-the-art algorithms.

Our main contributions can be summarized as follows:

- **Data Scarcity Solution:** We demonstrate that indirectly transferring knowledge from depth estimation significantly improves deblurring performance, providing a simple and practical solution to the limited availability of deblurring data without the need to significantly change the decoder part of the model.

- **Region-Adaptive Processing:** This work advances beyond AdaRevD by predicting blur levels for small 14×14 subregions rather than a single blur degree per 224×224 patch, enabling finer-grained adaptation. Moreover, the blur map is more deeply integrated into the network architecture, guiding processing directly instead of acting only as a stop gate as in AdaRevD.

- **State-of-the-art Results:** DRAMNet sets new state-of-the-art performance on both synthetic and real blur, demonstrating the practical value of our approach.

## 2 RELATED WORK

**Deblurring methods.** Early CNN-based methods like DeepDeblur (Nah et al., 2017b) and Sun et al. (2015) showed that using synthetically blurred frames, a cascade of per-scale feature extractors can effectively remove dynamic blur. This approach was streamlined by introducing a scale-recurrent network (Tao et al., 2018) that uses shared weights across pyramid levels, improving efficiency

without sacrificing quality. HINet (Chen et al., 2021b) further improved detailed reconstruction by applying instance normalization to channel groups, making training more stable.

GAN-based approaches, such as DeblurGAN (Kupyn et al., 2018), introduced adversarial training with a blur synthesizer and deblurring network to produce perceptually better results. An advancement was made by jointly training a blur generator (BlurGAN) and a deblurring network DB-GAN (Zhang et al., 2020), effectively narrowing the domain gap between synthetic and real-world blur.

Patch-hierarchical architectures address spatially varying blur by processing image patches at multiple scales; extensions like spatial attention modules (Suin et al., 2020a) or stacked convolutional networks like DMPHN (Zhang et al., 2019) adaptively focus on regions with more severe blur.

Multi-stage cascades, exemplified by a three-stage restoration pipeline MPRNet (Zamir et al., 2021), progressively refine deblurring with feature fusion between stages to capture both global and fine details. Scale-Recurrent Network (SRN) (Tao et al., 2018) and MT-RNN (Park et al., 2020) work with a similar concept utilizing lightweight pyramid block across resolutions. MIMO-UNet+ (Cho et al., 2021) generalizes the scheme to a multi-input multi-output formulation, and Whang et al.'s stochastic refinement network (Whang et al., 2022) samples candidate restorations and distills them, achieving strong accuracy.

Recently, transformer models like Uformer (Wang et al., 2022a) and Restormer(Zamir et al., 2022) have achieved state-of-the-art results by leveraging efficient self-attention blocks for global context. Stripformer (Tsai et al., 2022), DeepRFT+ (Mao et al., 2023), and MRLPFNet (Dong et al., 2023) progressively factorize or regularize attention in the frequency domain, UFPNet (Fang et al., 2023) adds a flow-based motion prior and trains a self-supervised model, showing that accurate kernel estimation can be learned even without paired data.

NAFNet64 (Chen et al., 2022) shows that a purely convolutional backbone with simple nonlinear activation can approach transformer quality if channels are allocated to different frequency paths.

Adaptive inference approach AdaRevD (Mao et al., 2024) uses region-specific blur severity to predict the number of decoder blocks for each region with a blur severity classifier, improving efficiency without sacrificing quality in complex areas. AdaRevD uniquely assesses regional blur but only for efficiency, not restoration guidance — gaps our work seeks to address.

**Priors in restoration.** Classical blind deconvolution relies on handcrafted image priors such as gradient sparsity or total variation, but modern deep architectures achieve far greater flexibility by learning powerful priors from data. The most common strategies are discussed in this subsection. Networks that operate in the Fourier or wavelet domain build an inductive bias toward periodic or edge-like structures. FFTformer (Yang et al., 2022) and the DWT $\rightarrow$ Conv $\rightarrow$ IDWT block (Suin et al., 2020b) are recent examples. A generator trained on natural images can act as a strong implicit prior: DeblurGAN (Kupyn et al., 2018) learns to invert motion blur via adversarial supervision. Large-scale pre-training on a geometry-related task injects structural knowledge that classic deblurring datasets lack. MiDaS (Ranftl et al., 2020) and DPT (Ranftl et al., 2021) showed that monocular depth generalizes across domains; Depth-Anything-v2 (Yang et al., 2024) scales this idea to 62M images.

Depth is particularly attractive for non-uniform deblurring: camera shake produces blur that varies with scene depth, while depth cues themselves remain relatively stable even in blurry frames; a depth-pretrained encoder therefore supplies dense, geometry-aware features that guide restoration far more effectively than signal-fidelity metrics such as PSNR. In our experiments these depth priors complement frequency experts and blur maps, yielding the largest quality gain and the most consistent convergence.

**Depth estimation methods.** Monocular depth estimation has progressed rapidly, moving from early encoder-decoder CNNs (Eigen et al., 2014; Laina et al., 2016; Fu et al., 2018) to large-scale pre-training (Godard et al., 2019; Johnston & Carneiro, 2020; Bhat et al., 2021) and vision transformers (Ranftl et al., 2021; Bhat et al., 2023). MiDaS (Ranftl et al., 2020) unified multiple depth data sets with scale-and-shift-invariant losses, while DPT (Ranftl et al., 2021) introduced a transformer backbone. Depth-Anything-v2 (Yang et al., 2024) scales training via teacher-student distillation, resulting in generic geometric priors that transfer well to downstream tasks.

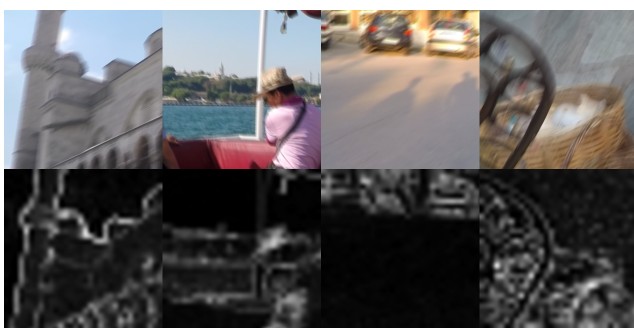

Figure 2: Examples of laplacian targets which we aim to restore blindly

**Datasets.** Although synthetic collections such as GoPro (Nah et al., 2017a) and REDS (Nah et al., 2019) provide tens of thousands of training pairs, they remain unable to capture the full complexity of real-world motion blur (Nah et al., 2017a; 2019). In contrast, RealBlur (Rim et al., 2020) and RSBlur (Rim et al., 2022) supply authentic blur examples but only on the order of a few thousand images each, creating a persistent gap in both scale and diversity. This imbalance directly underpins the data scarcity problem and the lack of representative, spatially varying blur distributions.

## 3   MODEL

Our DRAMNet model combines depth estimation pre-training, blur map estimation, and adaptive deblurring in a single network. The scheme of the method is shown in Figure 3a. The proposed model uses a shared *Encoder* module that extracts features from the blurred input. The features are propagated through the *Blur Map Estimation* module for the blur map. Both features and the blur map are fed through the *Deblurring Decoder* module to yield the restored image.

### 3.1   MODEL ARCHITECTURE

**Encoder.** We use a pretrained DINO-V2 encoder trained on the task of depth estimation (Yang et al., 2024) in order to transfer its geometrical prior knowledge to our task. We employ a two-stage training in order to adapt it to the deblurring domain. The details are listed below.

**Blur-map estimation.** To estimate the *local* strength of blur, we compute a per-region Laplacian blur map $B_{\text{pred}}$. For each $224{\times}224$ input patch, we apply four Blur-Map Estimation (BME) blocks to predict the absolute difference between the Laplacian responses of the blurred and ground-truth images, without using the ground-truth image as input. The structure of the BME block is listed in the subsection below.

Figure 2 visualizes initial images and their corresponding BME outputs: bright tiles correspond to strongly blurred areas, and dark tiles to nearly sharp regions. In contrast to AdaRevD's L1-based score, we estimate blur with a Laplacian-based map that targets perceptual blur severity in how blurred a patch appears to a human observer rather than how much signal fidelity is lost. The Laplacian accentuates the high-frequency content most suppressed by blur (Alutis et al., 2023) and correlates far better with perceived sharpness.

**BME block.** The blur map is produced by a cascade of four lightweight *Blur-Map Estimation (BME) blocks* attached to the shared encoder. Each BME block first refines its input feature map with two $3{\times}3$ Conv + BN + ReLU layers, then averages the activations over every non-overlapping $14{\times}14$ window and projects the result with a $1{\times}1$ convolution followed by sigmoid to obtain a low-resolution blur estimate $B_{\text{pred}} \in [0, 1]^{H_t \times W_t}$. A strided $3{\times}3$ Conv (stride 2) forwards a down-sampled feature map to the next BME stage, so the four blocks together cover the full $224 \times 224$ patch with a grid of $14{\times}14$ blur scores.

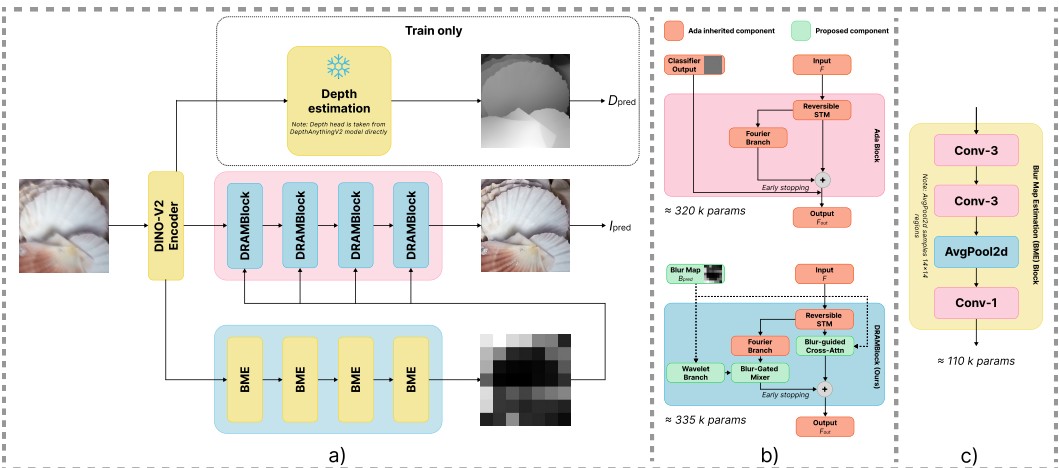

Figure 3: DRAMNet architecture. **(a)** Full pipeline: the shared encoder feeds a frozen Depth-Anything-v2 head, a multi-scale Blur-Map Estimation branch (BME), and a reversible decoder. **(b)** Internal layout of a DRAMBlock in comparison to AdaRevD block: the split-transform-merge (STM) core is followed by Fourier and blur-aware Wavelet experts, a blur-gated mixer, and a blur-guided cross-attention module. **(c)** Single BME block: two $3\times3$ convolutions refine features, average pooling over $14\times14$ windows yields a coarse blur estimate, and a strided convolution forwards features to the next scale.

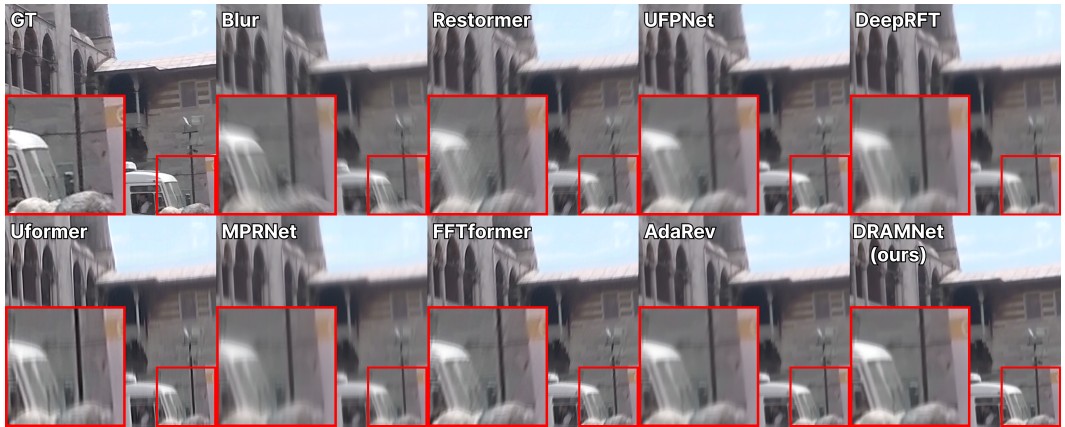

Figure 4: Comparison of deblurring results from various methods on one of the more challenging examples from GoPro dataset, including DRAMNet and AdaRevD-L. The bigger red-highlighted region is a zoomed-in crop taken from a smaller area to clearly illustrate the differences.

Formally, for each non-overlapping $14 \times 14$ tile $\mathcal{P}_{i,j}$ we compute:

$$B_{\text{Laplacian}}(i,j) = \frac{1}{|\mathcal{P}_{i,j}|} \sum_{(x,y)\in\mathcal{P}_{i,j}} \left|\Delta I_{\text{gt}}(x,y) - \Delta I_{\text{blur}}(x,y)\right|, \tag{1}$$

where $\Delta$ denotes the Laplacian operator, $|\mathcal{P}_{i,j}|$ is the element-wise module of the tile, $I_{\text{gt}}$ is the ground truth tile, and $I_{\text{blur}}$ is the blurred tile.

Each scale is supervised by the Laplacian-difference loss $\mathcal{L}_{\text{blur}} = \|B_{\text{pred}} - B_{\text{Laplacian}}\|_1$, ensuring consistent blur prediction across resolutions while adding only 0.44M parameters in total.

**Deblurring decoder.** Our decoder inherits the reversible Split-Transform-Merge (STM) core of AdaRevD but replaces the post-STM processing with a blur-adaptive DRAMBlock (Figure 3b) tailored to the coarse blur map available in our setting. After the encoder features $F_e^t$ traverse the unchanged STM core, they transform to the invertible tensor $F_{\text{rev}}^t$.

From this point the design diverges from Ada Block. We retain the original Fourier expert $\Phi_{\text{FFT}}$ and complement it with a blur-aware wavelet expert $\Phi_{\text{WAV}}$. Moreover, we use our blur map as additional guidance after the STM module via cross-attention.

The blur map is processed by a Wavelet branch. The input blur map $B_{\text{pred}}$ has $16 \times 16$ resolution (each pixel describes the blur level of a $14 \times 14$ image tile); a single bicubic up-sampling followed by a $3 \times 3$ smoothing filter converts it to the full feature resolution blur map $\tilde{B} \in [0, 1]^{1 \times H \times W}$. This refined map modulates the Haar DWT$\rightarrow 1 \times 1$-conv$\rightarrow$IDWT pipeline, amplifying high-frequency coefficients inside each tile in proportion to the local blur level and producing the wavelet feature $F_{\text{WAV}}^t$.

A blur-gated mixer then fuses the two experts. The first is a wavelet-processed blur map $F_{\text{WAV}}^t$. The second is an Ada-inherited Fourier branch $F_{\text{FFT}}^t$. Blur-gated mixer outputs their convex combination $F_{\text{mix}}^t = w_1 F_{\text{FFT}}^t + w_2 F_{\text{WAV}}^t$, thus allowing the model to find the balance between "pure Fourier" in sharp images and "wavelet boosted" in strongly blurred ones. $w_1$ and $w_2$ ($w_1 + w_2 = 1$) are estimated using a two-layer MLP.

In parallel, a lightweight blur-guided cross-attention head draws queries from $\text{LayerNorm}(F_{\text{rev}}^t)$ and uses a $1 \times 1$ projection of $\tilde{B}$ as keys and values, yielding the spatially selective feature $F_{\text{att}}^t$. The block output is the residual sum:

$$F_{\text{out}}^t = F_{\text{rev}}^t + F_{\text{mix}}^t + F_{\text{att}}^t, \tag{2}$$

which is similar to the vanilla Ada behavior when $B_{\text{pred}} \rightarrow 0$. As in AdaRevD, an early-exit rule skips the expert and attention paths whenever $\max B_{\text{pred}} < \tau$, preserving full invertibility and reducing the computational budget by roughly eight percent on lightly blurred patches.

### 3.2 LOSS CONSTRUCTION AND TRAINING PROTOCOL

The model is optimized in two stages. Stage 1 tunes the shared encoder to blurry input while keeping the geometric prior inherited from Depth-Anything-v2. Because GoPro lacks ground-truth depth, we generate a *pseudo-depth* target by running the original Depth-Anything-v2 model on the sharp reference frames. The encoder $E$ and the lightweight depth head $H$ are trained with the scale-invariant log loss

$$\mathcal{L}_{\text{depth}}^{(1)} \frac{1}{N} \sum_i \left( \log D_{\text{pred}}^{(i)} - \log D_{\text{pseudo}}^{(i)} \right)^2 - \frac{0.85}{N^2} \left( \sum_i \log D_{\text{pred}}^{(i)} - \log D_{\text{pseudo}}^{(i)} \right)^2, \tag{3}$$

where $N$ is the number of valid pixels, $D_{\text{pred}}$ is the depth predicted by $H$, and $D_{\text{pseudo}}$ is the pseudo-depth obtained from Depth-Anything-v2. After 60k iterations, the depth head is frozen and the loss is switched from pure depth loss to the weighted composite loss; the encoder continues to adapt during Stage 2.

Stage 2 fine-tunes the full pipeline end-to-end while keeping $H$ fixed. The objective is

$$\mathcal{L}_{\text{total}} = \mathcal{L}_{\text{recon}} + \lambda_d \mathcal{L}_{\text{depth}}^{(2)} + \lambda_p \mathcal{L}_{\text{perc}} + \lambda_b \mathcal{L}_{\text{blur}}, \tag{4}$$

where $I_{\text{out}}$ is the restored image, $I_{\text{gt}}$ is the sharp reference, and $\lambda_d, \lambda_p, \lambda_b$ weight the auxiliary terms. The reconstruction loss enforces pixel fidelity and sharp edges:

$$\mathcal{L}_{\text{recon}} = \|I_{\text{out}} - I_{\text{gt}}\|_1 + 0.5 \|\nabla I_{\text{out}} - \nabla I_{\text{gt}}\|_1, \tag{5}$$

where $\nabla$ is the spatial gradient. Depth consistency is supervised using the following equation:

$$\mathcal{L}_{\text{depth}}^{(2)} = 1 - \text{SSIM}(D_{\text{pred}}, D_{\text{pseudo}}), \tag{6}$$

that keeps the restored frame compatible with the frozen pseudo-depth. Perceptual loss is used to reduce over-smoothing of the model results:

$$\mathcal{L}_{\text{perc}} = \sum_{l \in \{3,8,15\}} \left\| \phi_l(I_{\text{out}}) - \phi_l(I_{\text{gt}}) \right\|_2^2, \tag{7}$$

where $\phi_l$ extracts VGG-19 features at layer $l$. Blur-map loss aligns the predicted blur map to its reference and supplies spatial guidance to every DRAM block:

$$\mathcal{L}_{\text{blur}} = \left\| \Psi_b(E(I_{\text{blur}})) - B_{\text{Laplacian}} \right\|_1, \tag{8}$$

Table 1: Quantitative comparison on four benchmarks. Best methods are highlighted in **bold**, second best are underlined

| Method | GoPro | | | REDS | | | RSBlur | | | RealBlur | | |
|---|---|---|---|---|---|---|---|---|---|---|---|---|
| | PSNR↑ | SSIM↑ | LPIPS↓ | PSNR↑ | SSIM↑ | LPIPS↓ | PSNR↑ | SSIM↑ | LPIPS↓ | PSNR↑ | SSIM↑ | LPIPS↓ |
| DeepDeblur | 29.1 | .914 | .185 | 27.6 | .868 | .200 | 25.7 | .804 | .270 | 32.5 | .841 | .185 |
| SRN | 30.3 | .934 | .160 | 28.8 | .887 | .175 | 26.8 | .822 | .240 | 35.7 | .947 | .150 |
| DMPHN | 31.2 | .940 | .145 | 29.6 | .893 | .158 | 27.7 | .835 | .228 | 35.7 | .948 | .148 |
| DBGAN | 31.1 | .942 | .150 | 29.6 | .895 | .162 | 27.6 | .833 | .235 | 33.8 | .909 | .160 |
| MT-RNN | 31.2 | .945 | .142 | 29.6 | .898 | .158 | 27.6 | .837 | .230 | 35.8 | .951 | .145 |
| MPRNet | 32.7 | .959 | .124 | 31.0 | .911 | .132 | 29.1 | .853 | .200 | 36.0 | .952 | .120 |
| HINet | 32.7 | .959 | .118 | 31.1 | .911 | .132 | 29.1 | .854 | .195 | — | — | — |
| MIMO-UNet+ | 32.5 | .957 | .122 | 30.8 | .909 | .138 | 28.9 | .850 | .205 | 35.6 | .947 | .122 |
| Whang et al. | 33.2 | .963 | .110 | 31.6 | .915 | .125 | 29.7 | .860 | .185 | — | — | — |
| Uformer | 33.1 | .967 | .105 | 31.4 | .919 | .118 | 29.5 | .858 | .180 | 36.2 | .956 | .105 |
| NAFNet64 | 33.7 | .967 | .100 | 32.0 | .919 | .110 | 30.0 | .863 | .170 | 35.8 | .952 | .100 |
| Stripformer | 33.1 | .962 | .108 | 31.4 | .914 | .120 | 29.5 | .857 | .178 | — | — | — |
| Restormer | 32.9 | .961 | .115 | 31.3 | .913 | .128 | 29.4 | .855 | .182 | 36.2 | .957 | .110 |
| DeepRFT+ | 33.5 | .965 | .112 | 31.8 | .917 | .125 | 29.8 | .861 | .178 | 36.1 | .955 | .108 |
| FFTformer | 34.2 | .968 | .098 | 32.5 | .920 | .108 | 30.5 | .867 | .163 | — | — | — |
| UFPNet | 34.1 | .968 | .102 | 32.4 | .919 | .115 | 30.3 | .866 | .175 | 36.3 | .953 | .095 |
| MRLPFNet | 34.0 | .968 | .103 | 32.3 | .919 | .117 | 30.3 | .865 | .178 | — | — | — |
| AdaRevD-B | 34.5 | .971 | .090 | 32.8 | .923 | .095 | 30.7 | .870 | .150 | **36.6** | .957 | .085 |
| AdaRevD-L | 34.6 | .972 | .088 | 32.8 | .924 | .093 | 30.8 | .871 | .148 | 36.5 | .957 | .083 |
| Ours | **34.8** | **.976** | **.082** | **33.4** | **.938** | **.083** | **33.8** | **.961** | **.128** | **36.6** | **.963** | **.075** |

where $B_{\text{Laplacian}}$ is the Laplacian-difference target.

The weights are fixed to $\lambda_d = 0.3$, $\lambda_p = 0.1$ and $\lambda_b = 0.2$; the ablation in Table 5 shows that this setting offers the best trade-off between PSNR, SSIM and LPIPS and remains stable under moderate perturbations.

### 3.3 IMPLEMENTATION DETAILS

All trainable parameters (except the frozen depth head) are optimized with AdamW (Loshchilov & Hutter, 2017) ($\beta_1 = 0.9, \beta_2 = 0.999$), starting at $3 \times 10^{-4}$ and decaying to $10^{-6}$ via a cosine schedule. Training runs in mixed precision on four A100 GPUs, processing $224 \times 224$ patches with batch size 32. End-to-end training on GoPro and RealBlur takes about 200 hours.

## 4 EXPERIMENTS

In this section, we describe the evaluation process of the proposed model on all popular real and synthetic deblurring datasets, comparing its performance with other methods. Additionally, we discuss the impact of depth pretraining and present the ablation studies conducted.

DRAMNet is trained on the GoPro and RealBlur training splits, and its performance is assessed on the held-out test sets of four datasets: GoPro, REDS, RSBlur, and RealBlur. We report results using the three most commonly adopted metrics in image deblurring benchmarks: peak signal-to-noise ratio (PSNR), structural similarity index (SSIM) (Wang et al., 2004), and learned perceptual image patch similarity (LPIPS) (Zhang et al., 2018a). These metrics are standardized across the literature and jointly capture different aspects of reconstruction quality: PSNR evaluates pixel-level accuracy, SSIM assesses structural similarity, and LPIPS measures perceptual closeness. Reporting all three ensures compatibility with existing evaluations and reflects a balanced view of restoration performance.

Table 2: Scene-wise performance on RSBlur test set

|  | Method | Indoor | Outdoor | Low-light | High-motion |
|---|---|---|---|---|---|
| **PSNR** ↑ | NAFNet64 | 32.90 | 32.70 | 31.10 | 32.20 |
|  | AdaRevD-L | 33.60 | 33.30 | 31.80 | 32.90 |
|  | Ours | **34.10** | **33.80** | **32.50** | **33.20** |
| **SSIM** ↑ | NAFNet64 | 0.937 | 0.934 | 0.917 | 0.929 |
|  | AdaRevD-L | 0.943 | 0.940 | 0.925 | 0.938 |
|  | Ours | **0.948** | **0.945** | **0.930** | **0.942** |

Table 3: Impact of depth pre-training on deblurring performance

| Initialization | GoPro | | | RSBlur | | |
|---|---|---|---|---|---|---|
|  | PSNR↑ | SSIM↑ | LPIPS↓ | PSNR↑ | SSIM↑ | LPIPS↓ |
| Random | 34.35 | 0.965 | 0.090 | 32.46 | 0.942 | 0.148 |
| DINO-v2 | 34.52 | 0.967 | 0.091 | 32.55 | 0.945 | 0.140 |
| Depth-Anything-v2 | **34.84** | **0.976** | **0.082** | **33.75** | **0.961** | **0.128** |

**Comparison with other methods.** Table 1 presents a comparison with other deblurring methods on four datasets. DRAMNet outperforms all prior methods on every benchmark. On GoPro and REDS we gain +0.24 dB and +0.55 dB in PSNR over the previous best; on RSBlur and RealBlur the improvements are +2.95 dB and +0.06 dB, confirming that depth-guided attention and blur map supervision enhance both synthetic and real-world deblurring without dataset-specific tuning.

To investigate performance across different types of scenes, we manually assigned each image from RSBlur datasets to one of four categories: indoor, outdoor, low-light, and high-motion. We then measured the PSNR and SSIM of the proposed DRAMNet and notable methods of NAFNet64 and AdaRevD-L under each category. The results are shown in Table 2.

DRAMNet achieves strong performance in all scenario types. High-motion scenarios are effectively handled because of the spatially varying blur processing. This localized adaptation helps prevent artifacts and preserves details across differently blurred areas. In low-light conditions, DRAMNet also benefits from adaptive filtering that enhances detail recovery.

Figure 4 demonstrates the qualitative results of our method and of other methods. DRAMNet is the most accurate image restoration method among the compared methods.

**Effect of depth pre-training.** To quantify the benefit of initializing our encoder and depth head with Depth-Anything-v2 weights, we compare DRAMNet against a variant where both components are randomly initialized and then trained end-to-end. To demonstrate that depth pretraining is the main reason for the high performance, not just the well-pretrained backbone, we also use DINO-v2 (Oquab et al., 2023) as the backbone and show that its results are worse than those of Depth-Anything-v2. The main reason for this is the strong correlation between an object's proximity to the camera and the amount of motion blur for moving objects; in other words, understanding the depth of the scene simplifies the motion blur estimation. Table 3 presents results on the GoPro and RSBlur test sets.

We also trained the model without the depth head; however, in the single-head setting, although the final metrics differed only slightly, the convergence and overall stability of the model were significantly worse.

Depth-Anything-v2 is trained on 595K synthetic labeled images and over 62M real unlabeled images using a teacher-student pseudo-labeling framework, which provides robust, fine-grained depth priors. It can be seen that the effect of depth pre-training is larger on RSBlur, where the real, scene-dependent blur is more complex than the synthetic motion blur and thus benefits more from the added structural priors.

Table 4: Depth pre-training (D) and blur-map estimation with DRAMNet Block (B) together improve baseline significantly. Standard deviation is measured for PSNR values over three runs

| D | B | PSNR ↑ | SSIM ↑ | Std. Dev. $\sigma$ ↓ |
|---|---|--------|--------|----------------------|
| ✗ | ✗ | 32.10 | 0.921 | $0.05 - 0.15$ dB |
| ✓ | ✗ | 32.21 | 0.933 | $0.05 - 0.15$ dB |
| ✗ | ✓ | 32.46 | 0.942 | $< 0.05$ dB |
| ✓ | ✓ | **33.75** | **0.961** | $< 0.05$ dB |

Table 5: Sensitivity of DRAMNet to loss weights on RSBlur

| $\lambda_d$ | $\lambda_p$ | $\lambda_b$ | PSNR ↑ | SSIM ↑ | LPIPS ↓ |
|-------------|-------------|-------------|--------|--------|---------|
| 0.3 | 0.1 | 0.2 | **33.75** | **0.961** | **0.128** |
| 0.1 | 0.1 | 0.2 | 33.20 | 0.955 | 0.140 |
| 0.5 | 0.1 | 0.2 | 33.50 | 0.959 | 0.135 |
| 0.3 | 0.2 | 0.2 | 33.45 | 0.960 | 0.130 |
| 0.3 | 0.1 | 0.1 | 33.30 | 0.958 | 0.142 |
| 0.3 | 0.1 | 0.3 | 33.25 | 0.957 | 0.132 |
| 0.0 | 0.0 | 1.0 | 32.65 | 0.948 | 0.155 |
| 0.0 | 0.1 | 0.3 | 32.90 | 0.952 | 0.148 |
| 0.3 | 0.0 | 0.2 | 33.33 | 0.957 | 0.140 |
| 0.3 | 0.1 | 0.0 | 32.55 | 0.946 | 0.160 |

**Effect of blur map estimation.** To quantify the contribution of blur map estimation, we train four variants of DRAMNet, toggling **(D)** the depth pretraining and **(B)** blur-map branch together with the replacement of every Ada block by a DRAM block. The resulting scores are reported in Table 4. Both depth pre-training and blur-map estimation separately already improve metrics, but when both components are enabled simultaneously, the improvement is much better. These results demonstrate that depth priors and blur-map supervision act synergistically: the depth prior is incorporated into BME, and, through a better understanding of the decoder's structure, it produces a more accurate assessment of blur in the patches.

**Loss components balancing.** To further examine our training setup, we performed a grid search on loss-weight hyperparameters $\lambda_d$, $\lambda_p$, and $\lambda_b$ around their default values $(0.3, 0.1, 0.2)$. As Table 5 shows, the selected combination yields the best overall trade-off of PSNR, SSIM, and LPIPS on RSBlur; deviations in any coefficient lead to modest drops in performance.

## 5 CONCLUSION

We have presented DRAMNet, a novel framework for single-image deblurring that tackles two core challenges: the scarcity of paired blur data and the non-uniform nature of real-world blur. First, by transferring structural priors from large-scale depth estimation datasets, our model learns robust features even when deblurring-specific training data are limited. Second, our region-adaptive blur-severity map guides the network to allocate more processing to heavily blurred areas while preserving details in sharper regions. Comprehensive experiments on both synthetic (GoPro, REDS) and real-world (RSBlur, RealBlur) benchmarks demonstrate that DRAMNet achieves state-of-the-art restoration quality across all metrics. Looking ahead, we aim to extend DRAMNet to video deblurring, leveraging temporal consistency for further gains, or alternatively to optimize the existing architecture for real-time inference, enabling practical deployment in live streaming scenarios.

## REPRODUCIBILITY STATEMENT

To ensure the reproducibility of our work, we have included the source code for the proposed model in the supplementary materials. We also included a detailed description of the training procedure and all the training hyper-parameters in the corresponding sections of the paper.

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

# 6 APPENDIX CONTENTS

This appendix provides additional details, experiments, and analyses to support the main paper. Below is a summary of each section:

- **Section 7:** Discussion of use-cases, limitations, and potential future improvements.
- **Section 8:** Additional visualizations for the model qualitative results.
- **Section 9:** Details on the blur map realization and reasoning.
- **Section 10:** Statistical confirmations of model improvements.
- **Section 11:** Breakdown on model performance in comparison to AdaRevD.

# 7 LIMITATIONS

We made several key design decisions in our approach, each with its own trade-offs.

First, we focus on deblurring single images rather than using a video-based model. We chose this simpler single-image approach to avoid the extra complexity of processing multiple video frames over time, especially since large-scale datasets of blurry videos are not widely available. The trade-off is that our model cannot leverage helpful information from neighboring video frames, so it might miss improvements that a dedicated video deblurring method could achieve. This is also the reason behind us not employing any optical flow in the model. However, extending our approach to the video domain remains an important direction for future work, and we plan to explore temporal consistency mechanisms and motion-aware modules to adapt DRAMNet to video deblurring settings.

Second, our model may still behave unexpectedly in certain real-world conditions or across different demographics due to domain bias. We have not extensively tested its performance on every possible group of people or type of scene (for example, very low-light settings or subjects with diverse skin

tones and ages). As a result, the model could show some bias or errors when faced with images that differ significantly from our training data. We plan further evaluation on more diverse, representative datasets and making any necessary refinements or adding safeguards before deploying the model in real-world applications.

Third, we incorporate a VGG-based perceptual loss during training and report the LPIPS metric, both of which rely on features from a pretrained VGG network. Because LPIPS is computed using the same backbone that guides our perceptual loss, our optimization may be implicitly tuned to that metric. Although this coupling can boost quantitative LPIPS scores, it reflects a common practice in perceptual image restoration and is not inherently detrimental.

Finally, in our experiments on RealBlur we only use the JPEG track (RealBlur-J). DRAMNet operates on standard sRGB images and relies on 8-bit gamma-corrected inputs, whereas the raw-sensor data in the RealBlur-R track requires demosaicing and color–space conversion steps that are beyond the scope of this work. Extending DRAMNet to operate directly on raw images would necessitate integrating a full ISP pipeline and is left for future research. Future work could investigate complementary or human-aligned perceptual criteria to provide a broader evaluation of visual quality.

## 8 VISUAL COMPARISON ON DIFFERENT SETS

Figure 5 compares the outputs of three state-of-the-art deblurring methods: DRAMNet, AdaRevD-L, and NAFNet64 on a variety of RSBlur test images. Across all examples, DRAMNet consistently delivers the most visually coherent results, effectively restoring fine structures and edge definitions while avoiding common artifacts.

In contrast, AdaRevD-L often leaves behind residual blur in regions of complex motion or texture. Although it reduces the overall blur, close inspection reveals that some mildly blurred areas remain underprocessed, leading to a slight softness compared to DRAMNet's outputs.

NAFNet64, despite its strong quantitative scores, exhibits noticeable visual inconsistencies. In particular, one can observe subtle banding and spurious high-frequency noise in areas that were originally smooth. These artifacts are not reflected in PSNR or SSIM metrics, highlighting the gap between numerical performance and perceptual quality.

In general, these comparisons demonstrate that DRAMNet's depth-aware priors and region-adaptive processing yield superior, artifact-free restorations across diverse real-world blur conditions, whereas competing methods may still suffer from under- or over-processing despite competitive metric values.

## 9 BLUR MAP

Figure 6 provides visualisations for both laplacian targets (second row) and blur map visualisations (third row).

Several works have explored the idea of predicting blur maps that estimate the spatial distribution and intensity of blur across an image. For example, Ma et al. (2018) propose end-to-end deep networks that produce dense binary blur maps using fully convolutional architectures. These models typically leverage high-level semantic features to distinguish between naturally smooth regions and genuinely blurred content. Others, such as Zhang et al. (2018b), incorporate attention mechanisms or multi-branch designs that jointly estimate the extent of blur and its perceptual desirability.

These works propose interesting ideas for extracting blur maps; however, they suffer from two key limitations. First, there is a lack of evidence regarding the correlation of the predicted blur maps with human perceptual judgment of blur severity. Second, the blur annotations used for supervision are based on heuristic assumptions and are inherently unstable, introducing inconsistency and bias into the training process.

In this work, we use a different approach to learning blur estimation by supervising the network with a physically meaningful target. Specifically, we define the training loss using the absolute difference between the Laplacian responses of the input (blurred) image and its corresponding sharp ground

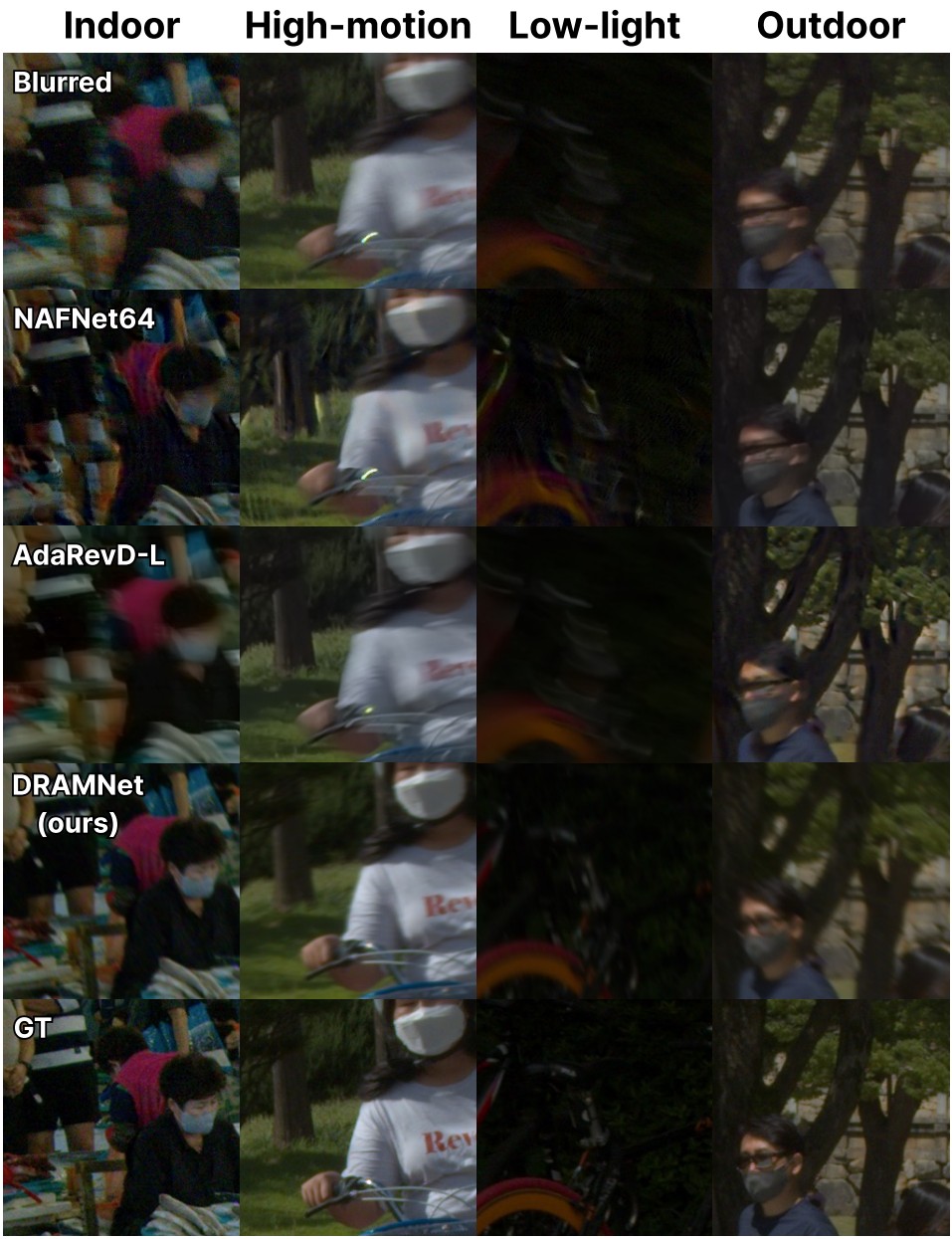

Figure 5: The comparison between several usecases from the ablation section of the main article

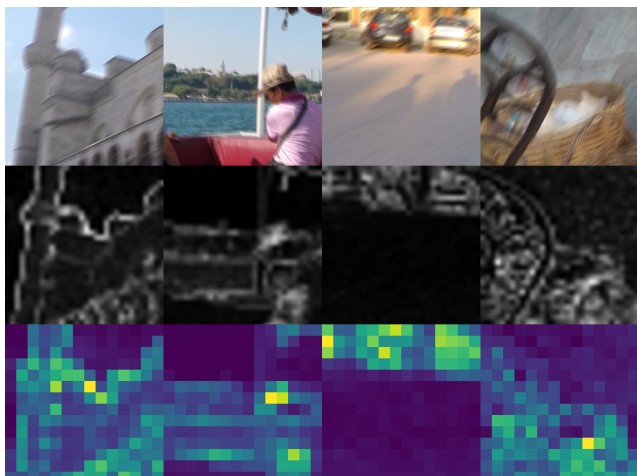

Figure 6: Examples of initial images containing blur, GT Laplassian targets and blur maps, produced by the BME block

| DB | ✓✓ | ✗✓ | ✓✗ | ✗✗ |
|----|------|------|------|------|
| ✓✓ | - | 0.0000 | 0.0000 | 0.0000 |
| ✗✓ | 1.0000 | - | 0.2499 | 0.0539 |
| ✓✗ | 1.0000 | 0.7501 | - | 0.0962 |
| ✗✗ | 1.0000 | 0.9461 | 0.9038 | - |

Table 6: P-values for pairwise comparisons (PSNR)

truth. This target encourages the model to focus on the loss of high-frequency details caused by blur, and provides a deterministic, interpretable signal for supervision.

Since the ground truth is not available at inference time, we train a lightweight network composed of repeated Blur-Map Estimation (BME) blocks to predict the blur map from the input image alone. This architecture allows us to decouple the interpretability of the supervision signal from the flexibility of the learned model. Compared to prior methods that regress human-annotated dense blur maps, our approach benefits from a well-defined and physically grounded loss and avoids reliance on subjective human annotations.

Among various candidates for defining blur supervision targets, we choose the Laplacian operator because of its strong theoretical and practical alignment with blur perception. As a second-order derivative filter, the Laplacian is highly sensitive to high-frequency content such as edges and fine textures, which is precisely the information that is most attenuated by blur. Unlike more complex metrics that require frequency transforms, structural templates, or learned components, the Laplacian is simple, computationally efficient, and fully interpretable. Empirically, it demonstrates consistently high correlation with human-perceived blur across multiple benchmarks Alutis et al. (2023). Furthermore, its linearity ensures a stable and convex loss surface when used as a regression target, making it particularly well-suited for training lightweight estimation networks. These properties make the Laplacian a robust and principled choice for constructing physically grounded blur maps.

## 10 STATISTICAL TESTS

We applied the one-sided Wilcoxon signed rank test to assess the statistical significance of comparisons between various parameters of our methods. This test is applicable because it is non-parametric and suited for paired samples without assuming normality. This test will show whether one parameter setup statistically outperforms another in terms of PSNR and SSIM. The results are provided in Table 6 and Table 7. D stands for depth pre-training and B stands for the blur-map branch, which uses the DRAMNet Block instead of the Ada Block.

| DB | ✓✓ | ✗✓ | ✓✗ | ✗✗ |
|---|---|---|---|---|
| ✓✓ | - | 0.0000 | 0.0000 | 0.0000 |
| ✗✓ | 1.0000 | - | 0.0397 | 0.0353 |
| ✓✗ | 1.0000 | 0.9603 | - | 0.3981 |
| ✗✗ | 1.0000 | 0.9647 | 0.6019 | - |

Table 7: P-values for pairwise comparisons (SSIM)

Table 8: Complexity vs. deblurring quality on GoPro ($256 \times 256$ input). DRAMNet adds a blur-aware decoder block and a four-stage BME head on top of AdaRevD-L, resulting in only a minor parameter overhead.

| Method | MACs (G) | Params (M) | Rel. MACs | PSNR (dB) |
|---|---|---|---|---|
| AdaRevD-L (UFPNet) | 460 | 210.8 | $1.00\times$ | 34.64 |
| DRAMNet (ours) | 485 | 211.7 | $\leq 1.06\times$ | **34.84** |
| *DRAMNet parameter breakdown* | | | | |
| Reversible decoder backbone | – | 107.8 | – | – |
| DRAMBlocks (32 blocks total) | – | 10.72 | – | – |
| Extra params vs Ada blocks | – | **+0.48** | – | – |
| Blur-Map Estimation head (4 blocks) | – | **0.44** | – | – |
| **Total DRAMNet params** | – | **211.7** | – | – |

To ensure reliability, we applied the Bonferroni correction, which controls the family-wise error rate across comparisons. This conservative adjustment minimizes false positives, reinforcing the significance of the results.

We used a significance level of $\alpha = 0.05$ for all statistical tests. The reported $p$-values correspond to one-sided Wilcoxon signed-rank tests. Comparisons with $p < \alpha$ are considered statistically significant. To account for multiple comparisons, we applied the Bonferroni correction, adjusting the threshold for significance to control the family-wise error rate. This correction ensures that the observed significance is not due to chance and confirms the robustness of our findings.

After correction, the comparisons involving the full model (D=✓, B=✓) remained statistically significant across both PSNR and SSIM metrics, indicating that the combination of depth pretraining and blur-awareness contributes meaningfully to performance gains.

## 11 PERFORMANCE

**Computational Cost.** As shown in Table 8, DRAMNet maintains nearly the same computational footprint as AdaRevD-L. Each DRAMBlock adds only $\approx$ 15k parameters over the original Ada block (335k vs. 320k), resulting in a total increase of $\approx$ 0.48M parameters across all 32 decoder blocks. The four-stage Blur-Map Estimation (BME) head adds an additional $\approx$ 0.44M parameters. Together, the full model increases total parameters by only $\approx$ 0.94M, i.e., a relative increase of just $\approx 0.4\%$ over AdaRevD-L.

Since MACs scale proportionally to block size and the BME modules operate on downsampled $14 \times 14$ features, the overall increase in computational cost is bounded by $\leq 6\%$. Importantly, DRAMNet preserves the reversible backbone, patch-wise processing, and early-exit mechanism of AdaRevD, so practical inference latency remains almost unchanged (typically within $5\%$).

