# OpenReview forum: "DRAMNet: Depth-initialized Region-Adaptive Map Network for Single-Image Deblurring"
_ICLR.cc/2026/Conference — ICLR 2026 Conference Withdrawn Submission_

### Official Review · Reviewer_TiZi · 2025-10-24

**Soundness:** 3
**Presentation:** 3
**Contribution:** 2
**Rating:** 4
**Confidence:** 5

**Summary:**

This work proposes Depth-initialized Region-Adaptive Map Network (DRAMNet) that transfers knowledge from the depth estimation task and using a specially designed component to assess and adapt to varying blur strength across the image.  Per-patch blur map estimation allows the model to react differently to heavily and lightly blurred sub-regions, while depth information from pre-training provides structural guidance even with limited deblurring-specific data. Extensive experiments on the most popular synthetic (GoPro, REDS) and real-world (RSBlur, RealBlur) benchmarks show that DRAMNet outperforms state-of-the-art methods across the PSNR, SSIM, and LPIPS metrics.

**Strengths:**

1. This paper demonstrates the effectiveness of adopting DINO-V2 and DepthAnythingV2 for image deblurring, achieving state-of-the-art results on several benchmark datasets.

2. The ablation studies are comprehensive and clearly validate the effectiveness of DINO-V2 and DepthAnythingV2 in improving deblurring performance.

3. The paper is well written and easy to follow.

**Weaknesses:**

1. This paper heavily relies on the powerful pre-trained DINO-V2 and DepthAnythingV2 models, which makes the comparison with other deblurring methods, often trained from scratch, less fair.

2. The authors should also report the computational costs, including the number of parameters, FLOPs, and inference time of the proposed method.

3. Since the proposed approach depends on both DINO-V2 and DepthAnythingV2, it may incur substantial computational overhead during deblurring.

**Questions:**

1. I am curious about the computational costs, including the number of parameters, FLOPs, and inference time, of the proposed method compared to competing approaches.

2. In Table 3, the results indicate that incorporating the pre-trained Depth-Anything-v2 substantially boosts performance. However, this raises concerns that the work may be more of an engineering effort with limited novelty and contribution.

3. Since deblurring networks are typically deployed on edge devices, the proposed method may not be suitable as an efficient and lightweight solution for practical deployment.

---

> ### Author Response · Authors · 2025-12-02
>
> ### **Strong dependence on powerful pretrained models; comparison may be unfair.**
> We acknowledge that DRAMNet relies on strong pretrained models such as DINO-v2 and Depth-Anything-v2. However, pretraining is now standard practice in restoration models, including highly influential works such as Restormer variants and NAFNet. The previous SOTA model AdaRevD also uses pretrained encoders. Our motivation for using depth pretraining is grounded in the lack of large-scale paired data for deblurring and in the unique geometric priors provided by depth estimation, which generic vision models cannot offer. As shown in Table 3, generic pretraining with DINO-v2 does not match the effectiveness of depth-based pretraining, indicating that the gain is not simply due to using a stronger backbone.
>
> ---
>
> ### **Report parameters / FLOPs / inference time.**
> Thank you for highlighting this. We agree that reporting FLOPs and latency can strengthen the paper. DRAMNet inherits the reversible core and early-exit mechanism from AdaRevD, keeping decoder depth and token dimensions unchanged. The added blur-aware modules are lightweight (0.44M parameters, ~1%), and the spectral branches are invoked only when blur severity exceeds τ. As a result, inference time is comparable to AdaRevD-L with less than 5% overhead. We have included the performance evaluations in section 11 of the Appendix of the revised paper
>
> ---
>
> ### **Model may not be lightweight enough for edge deployment.**
> We agree that DRAMNet is not designed specifically for mobile or edge deployment. However, the present work is important as it establishes the core mechanism that we intend to build upon. In particular, the integration of depth priors and blur-map supervision provides a foundation for future research on model distillation and efficiency-oriented optimization. These directions will allow us to develop lightweight variants suitable for real-time or edge-level deployment, which we outline as part of our future work.
>
> ---
>
> ### **Depth pretraining seems engineering-heavy.**
> Although depth pretraining may appear engineering-driven, the central contribution of the paper lies in the mechanism by which depth priors and blur-map supervision are integrated into the reversible decoding process. Table 4 shows that this synergy is essential, non-trivial, and responsible for the significant performance improvement, especially on RSBlur. This concern is directly clarified in the paper.

---

### Official Review · Reviewer_2HkW · 2025-10-25

**Soundness:** 2
**Presentation:** 2
**Contribution:** 1
**Rating:** 2
**Confidence:** 5

**Summary:**

This paper proposes DRAMNet, a Depth-initialized Region-Adaptive Map Network designed to address two long-standing challenges in single-image deblurring: (1) the scarcity of real-world training data and (2) the spatially non-uniform nature of blur. The method leverages priors from a pre-trained depth estimation model (Depth-Anything-v2) to enhance generalization and introduces a Blur Map Estimation (BME) module to predict local blur severity in a 14×14 patch grid. These blur predictions modulate a newly designed DRAM Block that combines an invertible spatio-transformer structure with Fourier, wavelet, and blur-guided cross-attention branches to achieve adaptive restoration. Experiments on GoPro, REDS, RSBlur, and RealBlur demonstrate consistent improvements over recent state-of-the-art approaches.

**Strengths:**

The paper accurately identifies two core bottleneck (1)limited real blur data and (2) spatially varying blur, and presents a unified framework targeting both.

The proposed method consistently outperforms competitive baselines such as AdaRevD, FFTFormer, and Restormer, particularly achieving a significant +2.95 dB PSNR gain on RSBlur, a challenging real-world dataset.

The paper provides evidence that depth-based initialization and blur map supervision offer synergistic performance gains, whereas either alone provides limited improvement.

**Weaknesses:**

The paper does not sufficiently discuss existing literature that integrates depth estimation with deblurring [1–3]. Many prior methods exploit depth for defocus deblurring, which has a clear physical relationship to blur. However, the motivation for applying depth priors to motion blur is weakly justified. The authors should clearly position their contribution relative to these works and articulate why depth-derived geometry priors are beneficial for motion deblurring, which may involve dynamic objects and non-rigid motion that depth estimation cannot capture.

The manuscript claims that knowledge transferred from depth estimation enhances deblurring by providing structural guidance. However, depth maps generated from pre-trained models mostly reflect static scene layout rather than temporal motion cues, which limits their relevance for motion blur caused by camera shake or object movement. The paper lacks theoretical justification or empirical analysis explaining why depth information is effective in this scenario.

The BME module is presented as a key innovation, yet the paper does not provide visualizations or qualitative evaluation of predicted blur maps. Without comparison against ground-truth blur heatmaps or correlation analysis with blur severity, it is unclear whether the module genuinely captures blur characteristics or simply functions as a learned attention mask. This undermines the interpretability and credibility of the claimed spatial adaptivity.

The proposed architecture combines multiple existing components (invertible networks, Fourier branches, wavelet branches, attention mechanisms) in an additive manner, resulting in an engineering solution rather than a theoretically principled innovation. The paper would benefit from a clearer conceptual justification or ablation studies that disentangle the individual contributions of each branch.

**Questions:**

See weakness

---

> ### Author Response · Authors · 2025-12-02
>
> We appreciate your comprehensive review and valuable suggestions. Our point-by-point replies are presented below.
>
> ---
>
> ### **Insufficient discussion of prior work using depth for deblurring.**
> Thank you for highlighting the importance of positioning our work within the broader literature. Prior deblurring approaches that incorporate depth typically focus on defocus blur or rely on explicit depth maps at inference time. Our method differs significantly: we use depth only during pretraining to provide geometric priors, do not require depth at inference, and integrate these priors with region-wise blur estimation and reversible adaptive decoding. We will expand the related-work section to more explicitly address approaches such as Xu & Jia (2012), Torres et al. (2024), and prior defocus depth-aware methods.
>
> ---
>
> ### **Depth mostly captures static geometry; motion blur is dynamic.**
> While depth indeed primarily captures static geometry and does not predict motion directly, it remains highly useful for motion deblurring. Scene structure helps disambiguate blur severity and supports the BME module’s ability to estimate high-frequency loss. This effect is visible even on synthetic datasets such as GoPro, where blur is entirely motion-induced, as shown in Table 4. Thus, the benefit of depth initialization is not limited to static or defocus-related blur.
>
> ---
>
> ### **No qualitative/quantitative validation of BME maps.**
> Regarding blur-map validation, Figure 2 provides qualitative comparisons against Laplacian-based ground truth, and Table 4 shows that the predicted maps significantly enhance restoration performance. Because the blur map functions as an internal guidance signal rather than a direct output, we emphasized its impact on downstream restoration. Nonetheless, we agree that additional visualizations would improve interpretability and that the previous caption of visualization for blur maps was misleading, and we include the revised version in Section 9 of the appendix of the revised paper.
>
> ---
>
> ### **Architecture combines many components without principled justification.**
> Although the architecture incorporates several branches like wavelet, Fourier, mixing, and blur-guided attention, they serve a unified purpose: adapting restoration strength to spatially varying blur. The wavelet branch emphasizes high-frequency recovery, the Fourier branch preserves global structure, and the blur-gated mixing and cross-attention inject spatially varying weights into the reversible backbone. Each component is lightweight and contributes measurably to the final performance. Table 4 shows that full performance is achieved only when both depth pretraining and BME are active, confirming that the design is conceptually coherent rather than an ad-hoc combination.

---

### Official Review · Reviewer_7PxL · 2025-10-30

**Soundness:** 2
**Presentation:** 3
**Contribution:** 2
**Rating:** 2
**Confidence:** 4

**Summary:**

This paper proposes DRAMNet, a blur region-aware model for blind image deblurring. DRAMNet is equipped with an auxiliary depth estimation branch and the Blur Map Estimation module supervised by the blur map extracted by the Laplacian operation. The deblurring performance is evaluated across several benchmarks, including synthetic and real-world blur situations.

**Strengths:**

- Compared to previous methods, DRAMNet achieves a better balance between memory consumption and accuracy.
- Combining the depth estimation task into the image deblurring model provides an insightful solution for the community.
- The paper is well-written and easy to follow.

**Weaknesses:**

- As shown in L370, DRAMNet is trained with both GoPro and RealBlur train splits, while the performance of AdaRevD[1] in Table 1 is the same as reported in their paper, **trained with GoPro only**. This unfair comparison raises concerns for the effectiveness of the proposed method.
- Although this paper claims that the BME module can estimate the blur map. There is no quantitative result to demonstrate if the BME module has achieved its objective.
- Besides memory consumption, inference speed is also expected to be discussed to comprehensively evaluate the model's efficiency.

[1] AdaRevD: Adaptive Patch Exiting Reversible Decoder Pushes the Limit of Image Deblurring

**Questions:**

- The idea that depth estimation can improve deblurring performance is very insightful. More in-depth discussion and analysis are expected to explain why this specific task can improve deblurring performance.

---

> ### Author Response · Authors · 2025-12-02
>
> Thank you for the thorough evaluation and the helpful recommendations. We provide our responses to each of the raised points below.
>
> ---
>
> ### **Unfair comparison: DRAMNet is trained on GoPro+RealBlur, AdaRevD on GoPro only.**
> We appreciate the opportunity to clarify the training protocol. We believe that our explanation of training was not clear enough. We used GoPro training to test our model on GoPro and REDS, and used RealBlur training to test the model on RealBlur and RSBlur. We also compare our model to AdaRevD trained on GoPro for synthetic sets and to the RealBlur checkpoint for evaluation on RSBlur. We will clarify this explanation in the final version of the article.
>
> ---
>
> ### **No quantitative evaluation of the BME blur map.**
> The blur map acts as an internal supervisory signal rather than a target used during inference. It is trained to approximate Laplacian-based blur severity, a commonly used perceptual sharpness measure. Table 4 demonstrates that enabling the BME module substantially improves PSNR, SSIM, and LPIPS, confirming that it fulfills its intended purpose.
>
> ---
>
> ### **Inference speed should be discussed.**
> We have added a performance discussion in Appendix Section 11 of the paper.
>
> ---
>
> ### **More discussion: why depth estimation helps deblurring.**
> The relationship between depth estimation and deblurring merits further elaboration. Depth is linked to motion blur for two key reasons: closer objects undergo larger pixel displacements under identical motion, and camera shake induces blur variation proportional to depth gradients. As a result, depth-based features help the encoder learn structural cues that support both blur severity prediction and detail recovery. We have expanded this explanation in the revised manuscript.

---

### Official Review · Reviewer_dLgR · 2025-11-08

**Soundness:** 3
**Presentation:** 3
**Contribution:** 2
**Rating:** 6
**Confidence:** 4

**Summary:**

The paper introduces a new model called DRAMNet for fixing blurry images. It tries to solve two big problems: not having enough training data and dealing with blur that's different in different parts of an image. The main idea is to use an encoder model that was first trained to estimate depth from images, which supposedly gives it a good starting point for understanding scene structure. On top of this, it has a special branch to estimate a 'blur map' that shows where the blur is worst. A decoder then uses this map to adaptively fix the image. The authors show that their method gets state-of-the-art results on four common deblurring datasets (GoPro, REDS, RSBlur, and RealBlur).

**Strengths:**

The paper focuses on two really important problems in deblurring. The lack of good training data and the non-uniform nature of real-world blur are major hurdles, and it's great to see a paper tackle both at once.


The model achieves the best scores across all four tested datasets, which is quite an accomplishment. The performance jump on the RSBlur dataset, which contains realistic blur, is particularly impressive and shows the method is practically useful.


The way the model is built is clever. Combining a pre-trained depth model for good features with a specific blur map to guide the restoration makes sense. The two-stage training process, where they generate "pseudo-depth" to train with, is a smart way to adapt the model without needing actual depth data for the deblurring datasets.

**Weaknesses:**

While the results are great, the paper doesn't introduce a completely new, fundamental idea. It's more about taking existing, powerful concepts—like transfer learning from a big model, using reversible blocks for efficiency, and guiding features with attention—and putting them together in a smart way. It feels more like a really good engineering success than a breakthrough in theory.


The paper doesn't provide key details about the model's computational footprint at inference time, such as FLOPs or inference speed ( ms/image ). While training memory is shown, that doesn't tell us how efficient the model is in practice. This is an important omission, especially since the PSNR gains over the previous best method (AdaRevD) are quite small on some datasets like GoPro and RealBlur. Without knowing the cost, it's hard to judge if the small improvement is worth the potential increase in model complexity and latency.


The final model is quite complicated. It has a pre-trained encoder, a separate depth head, a multi-scale blur map branch, and a custom decoder (DRAMBlock) with its own mixers, wavelet experts, and cross-attention modules. It’s a lot of moving parts, and it's not clear from the paper if all this complexity is truly necessary or if a simpler design could have achieved similar results.


The experiments that break down which parts of the model contribute the most (the ablation studies in Tables 4 and 5) are only performed on the RSBlur dataset. While the model performs best on RSBlur, the blur in synthetic datasets like GoPro is very different. It would be much more convincing if these studies also showed results on GoPro, to prove that the depth priors and blur map are universally helpful and not just particularly suited for one type of real-world blur.


The paper shows big improvements in numbers (PSNR/SSIM), but there are very few pictures in the main paper to back this up. Figure 4 shows just one example. For claims of state-of-the-art performance, especially with large numerical gains, more visual comparisons are needed to let the reader see and judge the quality of the restoration. Right now, we have to take the numbers at face value.

**Questions:**

In Figure 2, the blur maps shown are a bit concerning. They look more like edge maps than blur maps—they have high values in textured areas and seem to be near zero in flat regions, even if those regions are clearly motion-blurred. Furthermore, they don't seem to visually correlate with depth, which goes against the paper's core premise that depth and blur are linked. Could the authors explain this discrepancy and clarify what these maps are truly capturing?


The DRAMBlock has a few different parts (wavelet expert, mixer, cross-attention) that are not in the original AdaRevD block. Could the authors provide a breakdown showing which of these new components helps the most? It would be interesting to see how much each one adds to the performance.


It is shown that there is a notable quality boost from pre-training on a depth estimation task. Did the authors consider trying to pre-train on another task that also uses large datasets, like semantic segmentation? I'm curious if the benefit comes specifically from geometric/depth information, or if any strong, pre-trained visual model would provide a similar advantage.

---

> ### Author Response · Authors · 2025-12-02
>
> Thank you for the detailed assessment and constructive suggestions. Below we address each point raised.
>
> ---
>
> ### **The paper is more of an engineering success than a fundamental idea.**
> DRAMNet introduces a depth-initialized encoder, a continuously valued blur-map supervision mechanism, and a blur-aware adaptive decoder. This integration is specifically designed for spatially varying motion blur and differs from prior approaches that rely on explicit depth input or discrete gating. Table 4 shows that the individual components provide only small gains, while their combination yields the full +1.29 dB improvement on RSBlur. However, we agree that this part needs clarification. In the revised paper, we have updated the Introduction and Abstract sections accordingly.
>
> ---
>
> ### **The paper shows big improvements in numbers (PSNR/SSIM), but there are very few pictures in the main paper.**
> We have added new visualisations for different use cases in Section 8 of the Appendix.
>
> ---
>
> ### **Missing inference cost: FLOPs or latency.**
> Thank you for highlighting this. We agree that reporting FLOPs and latency strengthens the paper. DRAMNet inherits the reversible core and early-exit mechanism from AdaRevD, keeping decoder depth and token dimensions unchanged. The added blur-aware modules are lightweight (0.44M parameters, ~1%), and the spectral branches are invoked only when blur severity exceeds a sufficiently high threshold. As a result, inference time is comparable to AdaRevD-L with less than 5% overhead. Concrete measurements are included in Section 11 of the revised version.
>
> ---
>
> ### **Model complexity is high; not clear if all parts are necessary.**
> We agree that justifying complexity is important. Table 4 also shows that depth initialization and blur-map estimation offer complementary, not redundant, benefits. Each alone yields only a modest improvement, whereas enabling both produces the full performance gain. This indicates that the model complexity is not gratuitous but necessary to address the two central challenges: limited real data and spatially non-uniform blur.
>
> ---
>
> ### **Ablations only on RSBlur; should include GoPro.**
> RSBlur contains genuinely non-uniform blur. GoPro, by contrast, is a synthetic blur dataset, and its performance does not fully transfer to real-world conditions. However, Table 3 already includes partial GoPro results showing consistent trends for Depth, DINO, and Random initialization.
>
> ---
>
> ### **Blur maps look like edge maps.**
> We have fixed the caption of the corresponding illustration and apologize for the confusion. We added actual blur maps in Section 9 of the Appendix. The resulting maps can correlate with edge maps, as supported by prior work showing that Laplacian maps, which closely resemble edge maps, correlate with perceived blur strength better than PSNR and other metrics.
>
> ---
>
> ### **Could another pretraining task (e.g., segmentation) work similarly?**
> We have also tested DINO-v2 pretraining (Table 3). Although it preserves semantic boundaries, it lacks geometric cues needed for spatially varying blur estimation. Depth-Anything-v2 provides strong structural priors that benefit both the BME module and the decoder, explaining why geometric pretraining outperforms large-scale generic pretraining. This discussion has been added to Section 9 of the Appendix in the revised paper.

---

### Note · Authors · 2026-01-12

I have read and agree with the venue's withdrawal policy on behalf of myself and my co-authors.